# Spectrum-Weighted Fusion Cooperative Detection Algorithm Based on Double Thresholds for Underwater Acoustic Networks

**DOI:** 10.3390/s23167074

**Published:** 2023-08-10

**Authors:** Jing Zhang, Liyuan Lin, Rui Zhang

**Affiliations:** 1College of Electronic Information and Automation, Tianjin University of Science & Technology, Tianjin 300222, China; zjing@mail.tust.edu.cn (J.Z.); linly@tust.edu.cn (L.L.); 2College of Software and Communications, Tianjin Sino-German University of Applied Sciences, Tianjin 300350, China

**Keywords:** underwater acoustic communication, spectrum sensing, weighted fusion, energy detection

## Abstract

Spectrum-sensing technology is crucial for the development of underwater acoustic communication networks and plays a key role in detecting spectrum holes and channel occupancy. Energy detection technology, as the fundamental spectrum sensing technology in cognitive radio, has reached a mature level of development. Its application in hydroacoustic communications can significantly enhance the utilization of the hydroacoustic spectrum. However, due to the complexity of the hydroacoustic channel compared with that of the radio channel, the traditional double-threshold energy detection technique faces challenges such as fixed threshold values and limited flexibility. To address this, we propose a model for the hydroacoustic channel that incorporates a weight factor based on the signal-to-noise ratio in the algorithm. This allows for adaptive threshold values based on the user’s signal-to-noise environment, reducing false detection rates and improving overall detection performance. Through simulation experiments and comparisons, our proposed signal-to-noise weighted collaborative spectrum-sensing technique demonstrates superior detection performance compared with other spectrum-sensing techniques.

## 1. Introduction

The Earth’s surface is predominantly covered by oceans, accounting for approximately 71% or 360 million square kilometers [1]. The exploration and development of the ocean have spurred research on underwater communication technology [2,3]. Underwater acoustic sensor networks [4] are extensively used in marine exploration, disaster warning, sea area surveillance [5,6,7], etc. However, both artificial communication equipment and marine organisms utilize acoustic signals for communication, resulting in a scarcity of underwater communication spectra. Despite this, there are still some unused spectra that have not been fully utilized. Therefore, efficiently utilizing the underwater spectrum has become a key challenge in the development of underwater communication [8].

Cognitive radio technology was initially proposed to address the low spectrum utilization by cognitive users on land [9,10,11]. By detecting spectrum holes, sensing users enable cognitive users to quickly occupy unused spectra without interfering with authorized users. In 2007, Dr. Simon Haykin introduced this technology into underwater acoustic communication [12]. Cognitive radio technology helps address the scarcity of underwater spectrum resources and promotes the development of underwater communication technology. However, unlike land communication, radio waves experience significant attenuation underwater, with higher frequencies experiencing greater attenuation. This limits underwater communication to short-distance, high-speed communication, making long-distance underwater networking unachievable. Therefore, underwater acoustic communication primarily relies on acoustic wave transmission [13], with the communication channel constantly being influenced by natural conditions and random factors, resulting in complex and unstable physical properties [14,15]. Consequently, understanding and studying the underwater acoustic channel is the first step toward achieving underwater acoustic communication [16,17].

A comprehensive understanding of the current state and challenges of underwater communication and cognitive radio spectrum-sensing technology forms the basis for applying radio technology to underwater communication [18]. The energy-detection algorithm is the fundamental spectrum-sensing algorithm in cognitive radio technology. However, its performance can be affected by various uncertainties, such as fading channels [19], low sample size [20], and noise uncertainty [21]. Fadel F. Digham et al. [19] explored the no-diversity case and presented alternative closed-form expressions for the probability of detection (Pd). Luca Rugini et al. derived a novel, simple, and accurate analytical expression for the minimum number of samples required to achieve a desired probability of detection and false alarm rate by utilizing the cube-of-Gaussian approximation of chi-squared random variables [20]. Additionally, a new maximum likelihood (ML) estimation of noise and signal powers employing the cyclic prefix (CP) of orthogonal frequency division multiplexing (OFDM) was proposed to study the effects of unknown parameters on the energy detector [21]. Given that underwater nodes primarily rely on batteries, energy consumption must be limited, necessitating low-complexity algorithms [22].

The underwater spectrum is a scarce resource, and spectrum-sensing algorithms offer advantages over traditional algorithms for detecting underwater spectra. These challenges need to be addressed for the successful application of spectrum-sensing technology in underwater acoustic communication. While cognitive radio and cognitive hydroacoustic communication belong to different fields, the latter concept is derived from the former, suggesting a potential connection between the two. Some researchers have already suggested that the combination of these two technologies will help alleviate the spectrum resource constraints in hydroacoustic communication.

This paper is structured as follows: Section 2 provides an introduction and discussion of advanced research on underwater acoustic channels and energy-detection algorithms. Section 3 presents the underwater acoustic channel transmission model, including the calculation of the received signal, expression of the signal, and analysis of factors and effects that can impact the hydroacoustic signal in the hydroacoustic channel. In Section 4, the model and principle of traditional double-threshold detection and collaborative spectrum detection are presented. Section 5 analyzes the principle and flow chart of the algorithm. Section 6 compares the proposed algorithm with other traditional algorithms through simulation and discusses the limitations of each approach.

## 2. Related Work

In this section, we provide a detailed introduction to related works on spectrum-sensing technology. Currently, the majority of spectrum-sensing technologies are developed based on energy-detection algorithms, and they can be broadly categorized into three types:(1).Multichannel joint spectrum-sensing technologies [23]: In these technologies, the channel is divided into several subchannels for single-threshold energy detection, and the research focus is on optimizing the channel. The spectrum-sensing problem is treated as an optimization problem, where the goal is to maximize the system’s throughput. By exploiting the hidden convexity of the seemingly nonconvex problem, the optimal solution can be obtained. Subsequently, researchers began studying the problem of finding the optimal solution. In [24], the authors emphasized that ordinary mathematical tools can only optimize multiband aggregation throughput under convexity constraints. However, setting interference and utilization bounds limits the range of problems that can be solved using convexity. To address this, a genetic algorithm was proposed as an effective technique for solving nonconvex cooperative multiband sensing problems. In 2008, Xia Qiao-qiao et al. [25] identified some drawbacks of the genetic algorithm, such as a weak local search ability, premature convergence, slow late search, and the need for parameter adjustment. To overcome these limitations, the immune cloning algorithm (ICA) was introduced, which can quickly converge, maintain diversity, and suppress premature convergence. The algorithm considers both global and local search characteristics and relaxes the mathematical form of the objective and constraint functions. It is a simple and easy-to-implement algorithm suitable for solving nonlinear nonconvex complex system optimization problems. However, since most underwater nodes rely on batteries for power, energy savings should be prioritized.(2).Cooperative detection technologies [26,27,28,29] can be classified into two maincategories: (a) Centralized cooperative detection, where each sensing user reports their detection to a fusion center. The fusion center processes the information based on judgment criteria and distributes the results to each user. This approach allows the fusion center to effectively control the cognitive nodes involved in the collaboration. It also has lower requirements for detection accuracy, energy consumption, and computing power compared with single cognitive nodes. (b) Distributed cooperative detection, where each sensing user exchanges their detection information to determine accessible frequency bands. Multiuser communication traditionally uses eithera code-division multiple-access (CDMA) or time-division multiple-access scheme. The chirp signal resilience to multipaths and the Doppler effect makes it suitable for underwater acoustic communication [30]. The fusion modes for centralized cooperative detection fusion centers are as follows:
Hard merge: Each sensing user makes a decision based on the information. If a primary user is detected, the decision result is 1; otherwise, the decision result is 0, and the information is sent to the fusion center as 1-bit information [31].
1)AND criterion: All user decisions indicate the presence of the primary user, and the fusion center determines that the primary user exists.2)OR criterion: At least one sensing user believes that the primary user exists, and the fusion center determines that the primary user exists.3)K-out-of-N criterion.Soft merge: Each sensing user directly sends the detected signal to the fusion center. The fusion center performs fusion processing on the signals that were not processed by the sensing user. However, this approach requires significant expense and may reduce detection performance.Additionally, there are several specialized cooperative sensing algorithms, including the clustering-based cooperative spectrum detection algorithm [32,33], the joint spectrum detection algorithm based on relay [34], and the collaborative spectrum detection algorithm with limited bandwidth [35].(3).Multithreshold cooperative spectrum sensing involves the use of multiple thresholds and multiple users to enhance detection performance. In practical spectrum-sensing environments, the channel conditions are complex, which can impact detection effectiveness. Additionally, single-user detection can suffer from the hidden terminal problem and receiver interference due to obstacles. By enabling multiple sensing users to detect together and report their findings to a base station or fusion center, the detected information can be fused, and final decisions can be made based on specific decision rules. Multithreshold cooperative spectrum sensing not only improves the overall system detection performance but also reduces the accuracy requirement for single-user detection [36,37].

In low signal-to-noise ratio (SNR) environments, the energy statistics of authorized user signals and noise can be similar, leading to low detection probabilities with traditional single-threshold detection algorithms. To address this limitation, some studies have proposed a double-threshold detection model [38,39]. With this approach, unauthorized or secondary users (SUs) can dynamically switch between full-access mode and partial-access mode. By establishing a Markov chain model, the performance index of this strategy can be obtained, significantly improving the interference probability performance of the system [40].

Subsequently, researchers proposed combining the double threshold with cooperative detection [41] and weighting the energy values of the sensing users. They assigned relatively large weights to sensing users with high detection information SNRs while reducing the weights for deeply fading sensing users. This minimizes the adverse influence of detection statistics from deeply fading sensing users on the detection results [42], reduces cooperation costs, and improves detection performance. Quoc-Tuan Vien et al. [43] proposed a hybrid double-threshold-based energy detector (HDTED) scheme to enhance the performance of cooperative spectrum sensing (CSS) by leveraging both local decisions at SUs and global decision feedback from the fusion center (FC). However, in most double-threshold detection algorithms, the detection threshold is fixed, leading to poor detection effectiveness.

## 3. Underwater Acoustic Channel Transmission Model

The quality of communication is determined by the physical characteristics of the channel. In the case of underwater acoustic communication, the ocean medium presents certain characteristics such as a narrow channel band, a high noise level, and a multipath effect. The multipath effect is a significant challenge in underwater acoustic communication, as it refers to the existence of multiple propagation paths between the source and the receiver. As the signal propagates through the channel, the amplitude and phase change, resulting in a phenomenon known as fading. Rayleigh and Rice fading are examples of fading caused by the multipath effect. Therefore, when conducting channel simulation, it is crucial to fully consider these factors, assuming that sound waves in water are only reflected by the surface and bottom.

Furthermore, passing through a hydroacoustic channel leads to a loss of energy in the sound waves. This loss increases as the propagation distance grows, making propagation loss an important factor to be taken into account [44]. Propagation loss is typically expressed as *TL*:(1)$$\;TL=10log\frac{I_{1}}{I_{r}}$$

Assuming that the acoustic intensity per unit distance from the source is $I_{1}$, the acoustic intensity at distance *r* from the source is $I_{r}$.

In the actual ocean, the distribution of sound velocity is not uniform and is known to be influenced by temperature, salinity, and pressure according to Ude’s Equation (2).
(2)$$\;c=1450+421T-0.037T+1.14\left(S-35\right)+0.175P$$

In Equation (2), *T* represents temperature, *S* represents humidity, and *P* represents pressure.

A plan view of the speed of sound is shown in Figure 1, which shows that the speed of sound is affected to varying degrees during propagation as the depth of the ocean changes.

Assuming that the emitted sound source is located at a depth of 680 m, the sound line diagram is as shown in Figure 2.

The shallow-sea virtual source image is shown in Figure 3 [45], and the related parameters are shown in Table 1.

According to the known conditions, we can find the direct path *X* and the two paths *L*_1_ and *L*_2_, which are reflected only by the water surface. Finally, the total distance of the sound wave to the receiving end can be obtained, and then the time at which the sound wave reaches the receiving end can be determined:(3)$$\;\;\frac{X_{1}}{L_{1}}=\frac{X_{2}}{L_{2}}$$
(4)$$\;\sqrt{L_{1}{}^{2}-X_{1}{}^{2}}+\sqrt{L_{2}{}^{2}-X_{2}{}^{2}}=D$$

According to Equations (3) and (4), the expressions of *L*_1_ and *L*_2_ can be obtained.

According to this calculation method, the total distance traversed by the sound waves reflecting twice can be calculated as follows:(5)$$\;\;L_{1}{}^{2}={\left(\frac{X_{1}\times D}{X_{1}+X_{2}}\right)}^{2}+X_{1}{}^{2}$$
(6)$$\;\;\;L_{2}{}^{2}={\left(\frac{X_{1}\times D}{X_{1}+X_{2}}\right)}^{2}+X_{2}{}^{2}$$

If *S(t)* is the signal emitted by the sound source, considering that the acoustic wave will produce a certain attenuation and delay in the channel, the received waveform can be expressed as
(7)$$\;\;S\left(t\right)=A_{0}S\left(t-\tau_{0}\right)+\overset{y-1}{\underset{i=1}{\sum }}A_{i}S\left(t-\tau_{i}\right)+n\left(t\right)$$
where *y* represents the number of paths, *A*_0_ and *A*_i_ are the amplitudes through the receiving point, *τ_0_* and *τ*_i_ denote the delays through the receiving point, the first term is the direct wave, the second term is the reflected sound wave, and the third term is the interference noise.

## 4. Double-Threshold Cooperative Energy-Detection Model for Underwater Acoustic Communication

This section introduces double-threshold energy detection and cooperative spectrum detection.

### Double-Threshold Energy-Detection Model and Principle

The energy-detection algorithm is a traditional detection method that is simple to implement and does not require prior knowledge of the signal. First, a single threshold is adopted. The basic detection model is shown in Figure 4.
(8)$$\;\;Y\left(t\right)=\left\{\begin{array}{c} \begin{array}{c} s\left(t\right)+n\left(t\right)\;\;\;\;\;\;\;\;H_{1}\;\\ \;n\left(t\right)\;\;\;\;\;\;\;\;\;\;\;\;\;\;\;\;\;H_{0}\; \end{array} \end{array}\right\}$$

In Equation (8), *Y(t)* represents the received signal, *s(t)* is the user’s signal, *n(t)* denotes the Gaussian white noise signal, *H*_0_ means that the frequency band is not occupied, and *H*_1_ means that the frequency band is not utilized.
(9)$$\;\;E=\frac{1}{M}\overset{M}{\underset{m=1}{\sum }}{\left|Y\left(t\right)\right|}^{2}$$

Equation (9) represents the energy value of the received signal, where *M* is the total number of sampling points of the signal. When the number of sampling points is sufficiently large, the energy value of the received signal follows a Gaussian distribution.
(10)$$\;\;\;\;\left\{\begin{array}{c} E\sim Normal\left(M\sigma_{n}^{2}\;,\;2M\sigma_{n}^{4}\right)\\ E\sim Normal\left\{M(\sigma_{n}^{2}+\sigma_{S}^{2}),2M{\left(\sigma_{n}^{2}+\sigma_{S}^{2}\right)}^{2}\right\} \end{array}\right\}$$

In Equation (10), $\sigma_{n}^{2}$ represents the signal variance, and $\sigma_{s}^{2}$ represents the noise variance. Therefore, the false alarm probability of the received signal can be expressed as
(11)$$p_{f}=P\left\{E>T|H_{0}\right\}=Q\left(\frac{T-M\sigma_{n}^{2}}{\sqrt{2M\sigma_{n}^{4}}}\right)$$

In the case of a constant false alarm probability, $p_{f}$ is known, and the threshold *T* can be obtained from Equation (11), where $Q\left(X\right)=\frac{1}{\sqrt{2\pi }}\int_{X}^{\infty }e^{-\frac{2}{t^{2}}}dt$ is a Gaussian complementary cumulative distribution function:(12)$$\;\;\;T=Q^{-1}\left(P_{f}\right)\sqrt{2M\sigma_{n}^{4}}+M\sigma_{n}^{2}$$
(13)$$\;p_{d}=P\left\{E>T|H_{1}\right\}=Q\left(\frac{Th-M(\sigma_{n}^{2}+\sigma_{S}^{2})}{\sqrt{2M{\left(\sigma_{n}^{2}+\sigma_{S}^{2}\right)}^{2}}}\right)$$

The detection probability $p_{d}$ can be calculated by substituting the calculated threshold T into Equation (13) and using the Monte Carlo method for simulation.

The double-threshold energy-detection model is shown in Figure 5, and the signal energy values follow a Gaussian distribution. If the energy value of the received signal is greater than *T*_2_, the detector deduces that the primary user exists; if the energy value of the received signal is less than *T*_1_, the detector deduces that the primary user does not exist. If the energy value of the signal is between the thresholds, detection is performed again. If the detection result does not change, this part of the sensing data is directly eliminated, saving the expense caused by data uploading and processing.
(14)$$\;p_{f}=P\left\{Y>T_{1}|H_{0}\right\}=Q\left(\frac{T_{2}-M\sigma_{n}^{2}}{\sqrt{2M\sigma_{n}^{4}}}\right)$$
(15)$$p_{m}=1-p_{d}=1-Q\left(\frac{T_{1}-M(\sigma_{n}^{2}+\sigma_{S}^{2})}{\sqrt{2M{\left(\sigma_{n}^{2}+\sigma_{S}^{2}\right)}^{2}}}\right)$$

Thus, the expressions of the relationship between the missed detection probability $p_{m}$ and false alarm probability $p_{f}$ and the two threshold values *T*_1_ and *T*_2_ can be obtained, and the threshold values can be deduced as follows:(16)$$T_{1}=Q^{-1}\left(1-p_{m}\right)\sqrt{2M{\left(\sigma_{n}^{2}+\sigma_{S}^{2}\right)}^{2}}+M(\sigma_{n}^{2}+\sigma_{S}^{2})$$
(17)$$\;\;T_{2}=Q^{-1}\left(p_{f}\right)\sqrt{2M\sigma_{n}^{4}}+M\sigma_{n}^{2}$$

## 5. Weighted Double-Threshold Cooperative Spectrum–Sensing Algorithm

The signal-to-noise ratio (SNR) is a crucial metric for assessing the quality of transmission. A higher SNR indicates better transmission quality. The SNR is defined as the ratio of signal energy to noise energy and is typically expressed as
(18)$$SNR=\frac{I_{s}}{I_{n}}$$

$I_{s}$ represents the signal energy, and $I_{n}$ represents the noise energy.

The emission source level $s_{l}$, the propagation loss level $\;p_{l}$, and the noise level $n_{l}$ are denoted as [44]
(19)$$s_{l}=\frac{I}{I_{0}}$$
(20)$$\;p_{l}=\frac{I}{I_{s}}$$
(21)$$n_{l}=\frac{I_{N}}{I_{0}}$$

I is the sound intensity of the acoustic wave, and $I_{0}$ is the reference sound intensity.
(22)$$I_{0}=\frac{p_{0}^{2}}{Z_{0}\left(x\right)}$$

$p_{0}$ is the reference sound pressure, and $Z_{0}\left(x\right)$ is the acoustic impedance of the medium.

Thus, the signal-to-noise ratio can also be expressed as
(23)$$SNR=\frac{n_{l}}{s_{l}*p_{l}}$$

Equation (23) shows that the signal-to-noise ratio is position dependent.

The proposed algorithm incorporates an SNR weight factor, which enables dynamic adjustment of the two threshold values based on the SNR conditions, resulting in improved overall detection performance. Figure 6 illustrates the algorithm flow for each user to perform double-threshold detection.

For the actual sea area, it is assumed that the sound velocity c, the transmitting sound source, the sea depths X_1_ and X_2_ of the transmitting/receiving point in the marine environment, and the distance D between the transmitting/receiving points are known. According to ray theory, if there are y intrinsic sound lines, there are y paths to the receiving end. The received signal is obtained by superimposing Gaussian white noise. According to Equation (9), the energy value of the received signal can be calculated.According to Equations (14) and (15), the high and low thresholds *T*_1_ and *T*_2_ can be calculated by the known false alarm probability and the missed detection probability.The weight factor is calculated according to the SNR of each user, and the formula is as follows:(24)$$\;\rho_{i}=\frac{SNR_{i}}{\frac{1}{L}\sum_{i=1}^{L}SNR_{i}}$$
where $\;\rho_{i}$ is the weight factor corresponding to the ith cognitive user, and L is the number of cognitive users.The threshold value weighted by the SNR can be expressed by Equations (25) and (26). In an environment with a low SNR, the threshold value will increase, while in an environment with a high SNR, the threshold value will decrease.The calculated energy value of the received signal is compared with the high threshold $T_{2}^{i}$. If the energy value is higher than the threshold, the signal is judged as the primary user. If the energy value is lower than this threshold, the signal will enter the next cycle, and the energy value will be compared with the lower threshold $\;T_{1}^{i}$. If the energy value is lower than this threshold, it will be judged that there is no primary user. If the energy value of the signal is between the thresholds, it will be redetected. If there is no change in the final result, this part of the data will be discarded. The detection probability can be expressed as $T_{2}^{i}$ and $\;T_{1}^{i}$:(25)$$\;T_{1}^{i}=\frac{T_{1}}{\rho_{i}}$$
(26)$$\;T_{2}^{i}=\frac{T_{2}}{\rho_{i}}$$
(27)$$p_{d}^{i}=P\left\{E>T_{2}^{i}|H_{1}\right\}=Q\left(\frac{T_{2}^{i}-M(\sigma_{n}^{2}+\sigma_{S}^{2})}{\sqrt{2M{\left(\sigma_{n}^{2}+\sigma_{S}^{2}\right)}^{2}}}\right)$$The fusion center makes a decision according to the detection results of each cognitive user with the OR criterion and adopts the Monte Carlo simulation N_simu = 1000 times.

## 6. Results and Discussion

In the simulation of the shallow-sea hydroacoustic channel, a ray acoustic model is employed. This model assumes that the acoustic wave undergoes multiple surface and bottom reflections during propagation, resulting in the receiving point being exposed to several wavefronts propagating along different paths.

Figure 7 illustrates the amplitude and frequency characteristics of the hydroacoustic signal. Equation (7) describes the received signal, which consists of the direct path and reflected paths. The main frequency components of the signal are depicted in Figure 7. To analyze the signal in the frequency domain, the fast Fourier transform (FFT) is utilized to convert the time-domain signal into the frequency domain. The Fourier transform is a fundamental algorithm in digital signal processing. It is based on the principle that any time sequence or continuous measurement signal can be represented as a combination of sinusoidal signals with different frequencies. The spectrum of the signal can be extracted using the Fourier transform, facilitating subsequent spectrum analysis. In the amplitude and frequency characteristic graph, the two peaks represent the primary frequency components of the hydroacoustic signal in the frequency domain, while the spiked portion at the bottom represents white noise.

To verify the effectiveness of the proposed algorithm, a simulation of the algorithm is presented below, setting $p_{f}$ = 1 × 10^−6^ and $p_{m}$ = 0.01. The variance of white Gaussian noise $\sigma_{n}^{2}$ is 10. In the simulation experiment, we first compare the proposed algorithm with three traditional spectrum-sensing algorithms, and the simulation results are shown in Figure 8. Then, simulation experiments are performed on the collaborative detection algorithm using the K-judgment criterion, as shown in Figure 9, and the reason why the K-judgment criterion is worse than the OR-judgment criterion is also determined. Finally, a simulation comparison is performed between the improved algorithm and the original algorithm to verify the feasibility of the proposed algorithm, as shown in Figure 10.

In the simulation, four algorithms are compared under the condition of a constant false alarm rate. These algorithms include the following:(a)Single-user single-threshold detection: This represents the traditional energy-detection algorithm.(b)Single-user double-threshold detection: This algorithm incorporates two threshold values for improved detection performance.(c)Double-threshold cooperative detection algorithm: This algorithm utilizes the OR criterion at the fusion center for collaborative detection.(d)Weighted double-threshold collaborative detection algorithm: This is the algorithm proposed in this paper, which incorporates a weighted factor for dynamic adjustment of the threshold values.

Figure 8 demonstrates the detection performance of these algorithms. It can be observed that the basic single-user single-threshold detection algorithm has the poorest performance. The single-user double-threshold energy-detection algorithm outperforms the single-user single-threshold algorithm, and the collaborative detection algorithm outperforms the single-user double-threshold energy-detection algorithm.

Figure 8 further illustrates that the detection probabilities of the weighted double-threshold collaborative detection algorithm and the two-threshold collaborative detection algorithm, at an SNR of −7 dB, are already close to 1, with values of 0.972 and 0.979, respectively. In contrast, the detection probabilities of the remaining two algorithms at this SNR are only 0.643 and 0.796, respectively, which are significantly lower. Even at the highest SNR, the single-user single-threshold energy-detection algorithm and the single-user double-threshold energy-detection algorithm do not achieve a detection probability of 1.0. However, the double-threshold algorithm performs better than the single-threshold algorithm. This is because at relatively low SNRs, the signal and noise powers are comparable, making detection more challenging. The double-threshold algorithm can more accurately detect the frequency band state, resulting in a detection probability approaching 1 at −1 dB.

Overall, the fourth algorithm, the weighted double-threshold collaborative detection algorithm, exhibits the best detection performance. The simulation comparison diagram provides a visual representation of these results.

In this paper, we propose a centralized collaborative detection algorithm, where the sensing information from each user is fused at the fusion center. The fusion center then processes the received information using various fusion judgment algorithms, including the AND criterion, the OR criterion, and the K-out-of-N criterion. We simulate and analyze the fusion judgment criteria for the collaborative detection algorithm.

The K-out-of-N criterion implies that k users are identified as being in the vicinity of the main user, and the final judgment result is that the band is occupied. The value of k can be adjusted based on the specific requirements. In contrast, the AND criterion requires all users to have the same perception results. However, in practical scenarios, it is not guaranteed that all perceived users are in the same environment, which introduces uncertainty in the final detection results. Therefore, we do not conduct simulation experiments on the fusion judgment algorithm in this section.

Figure 9 presents the simulation results of the K-out-of-N criterion for different values of k, specifically 3, 4, and 5. From Figure 10, it can be observed that the overall effects of the three curves are similar, and the detection probability reaches 1 in the −5 dB environment. With a total of 5 users, setting k to 5 is equivalent to the AND criterion. However, in the −15 dB to −10 dB interval, the curve fluctuates and remains in an unstable state. Although the K-out-of-N criterion allows for adjusting the value of k to achieve the best effect, if the number of users is large, constantly updating and adjusting the k value can be labor-intensive and time-consuming.

The OR fusion criterion is considered more suitable among the fusion judgment algorithms because it does not require the same environment for all users as in the AND criterion, and it avoids the constant adjustment of the k value required by the K-out-of-N criterion. This algorithm addresses the limitations of the double-threshold centralized collaborative detection algorithm in low signal-to-noise ratio scenarios. The fixed threshold values in the double-threshold algorithm are inflexible and result in weak detection performance. By introducing a threshold factor that can be adjusted based on the signal-to-noise ratio, the flexibility of the threshold is improved, leading to enhanced overall detection performance.

The simulation comparison diagram in Figure 10 demonstrates that the proposed algorithm addresses the issues present in traditional collaborative detection algorithms and improves the overall effect. For instance, when the signal-to-noise ratio is −20 dB and the number of users is the same, the simulation curves of the proposed algorithm and the traditional double-threshold OR criterion collaborative detection algorithm have values of 0.64 and 0.472, respectively, indicating the superior performance of the proposed algorithm. Moreover, the proposed weighted double-threshold OR criterion collaborative detection algorithm tends to achieve stability earlier, indicating a more stable detection effect. Although the OR criterion collaborative detection algorithm does not reach the maximum detection probability, the proposed algorithm achieves a detection probability of 1 under the condition of −1 dB, demonstrating the effectiveness of the proposed algorithm.

## 7. Conclusions

Energy-detection algorithms are widely studied in the field of cognitive radio due to their versatility in detecting various signal types and their simplicity. However, as communication technologies continue to advance, single-threshold energy-detection algorithms are no longer sufficient to meet user requirements. Consequently, various spectrum-sensing algorithms have been developed to address this issue. This study focuses on spectrum-sensing techniques in underwater acoustic communication and introduces cognitive radio spectrum-sensing technology.

Considering that the propagation of hydroacoustic signals in water significantly differs from that of radio signals and is affected by factors such as attenuation and time delay, we have developed a channel model. Building upon a traditional energy-detection algorithm, we propose the use of a weighting factor that allows for dynamic adjustment of the double threshold based on the signal-to-noise ratios of individual users. This enhances flexibility and improves the detection probability. We compare the results of our proposed algorithm with those of different algorithms under various conditions, demonstrating its effectiveness. Through simulation experiments, we compare our proposed double-threshold collaborative detection algorithm with several existing perception algorithms, showing that it outperforms them with better results.

Furthermore, we conduct simulation experiments to compare the fusion judgment algorithms used in the fusion center of the collaborative detection algorithm. The results of these experiments confirm the significant impact of the algorithm proposed in this paper on hydroacoustic spectrum perception.

## Figures and Tables

**Figure 1 sensors-23-07074-f001:**
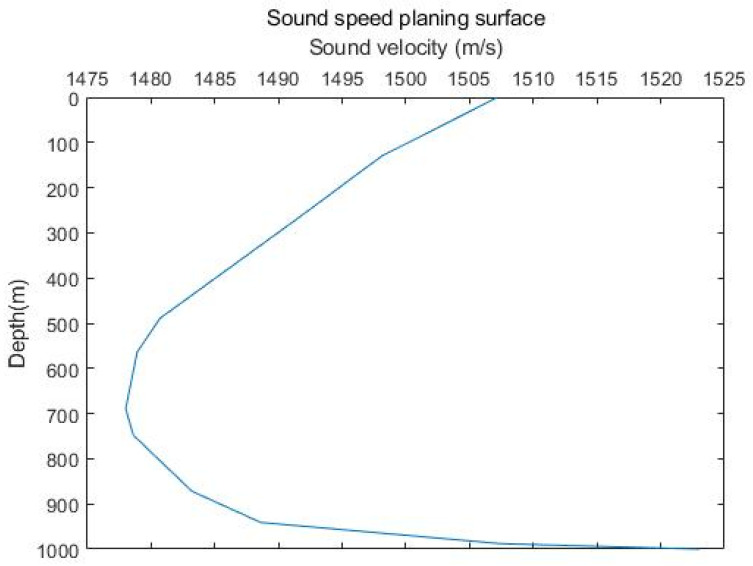
Planar surface of the speed of sound.

**Figure 2 sensors-23-07074-f002:**
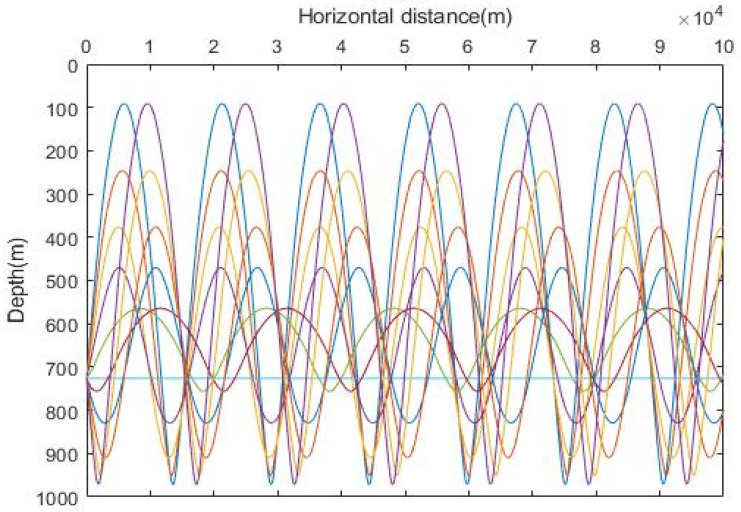
Sound line trajectory diagram. Different colored lines represent different paths of sound lines.

**Figure 3 sensors-23-07074-f003:**
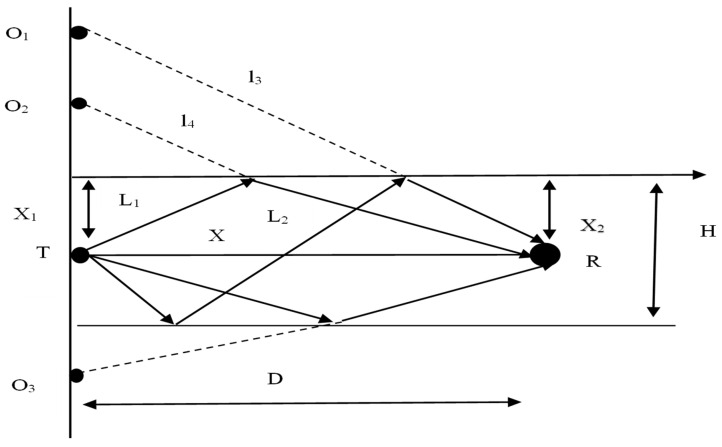
Shallow-sea virtual source image.

**Figure 4 sensors-23-07074-f004:**
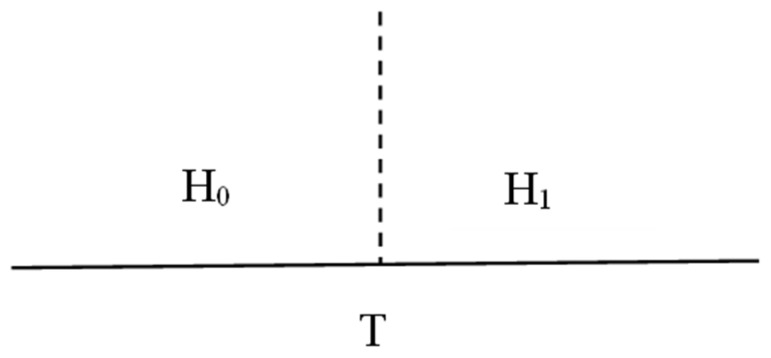
Model diagram of single-threshold detection.

**Figure 5 sensors-23-07074-f005:**
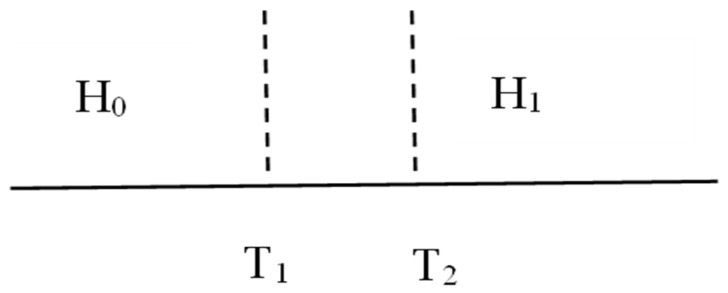
Model diagram of double-threshold detection.

**Figure 6 sensors-23-07074-f006:**
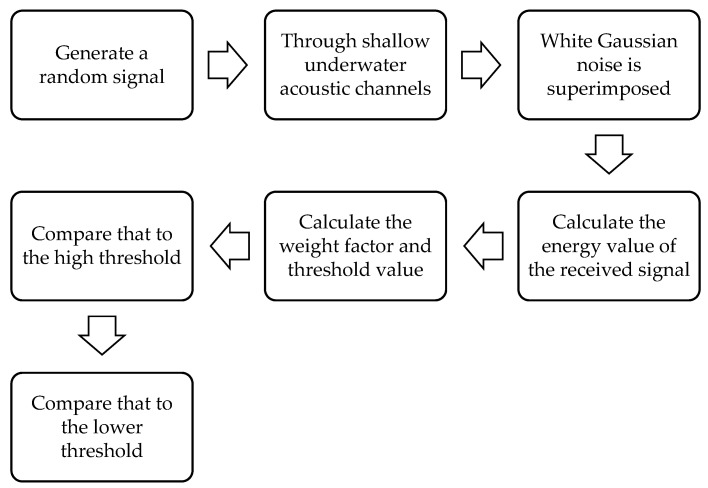
Detection flow chart.

**Figure 7 sensors-23-07074-f007:**
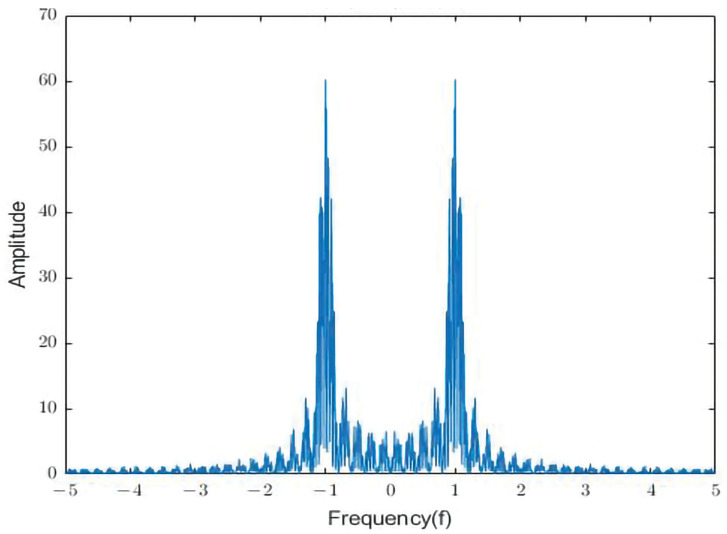
Amplitude and frequency characteristic curve of the hydroacoustic signal.

**Figure 8 sensors-23-07074-f008:**
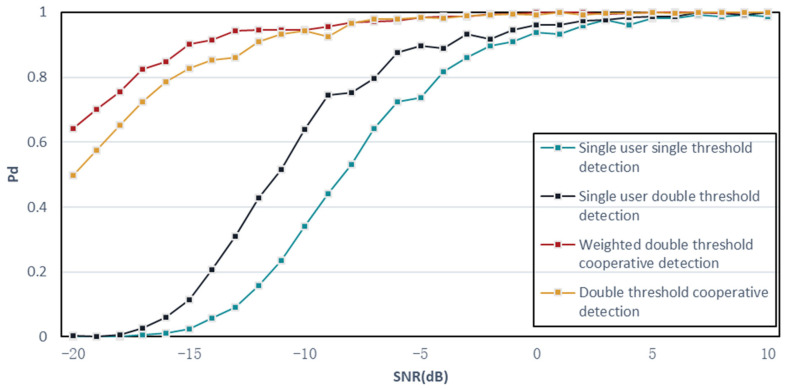
Detection algorithms simulation experiment comparison chart.

**Figure 9 sensors-23-07074-f009:**
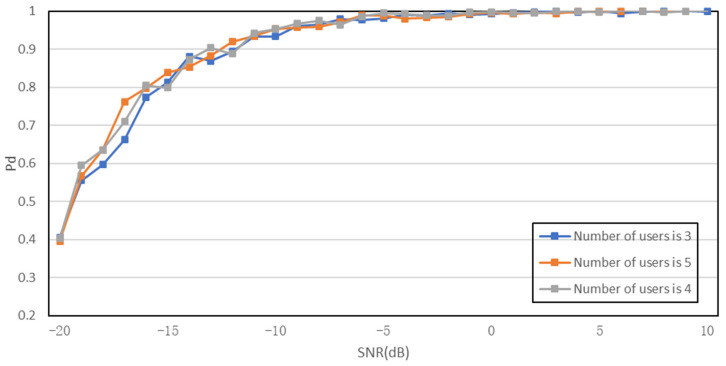
Comparison chart of K-criteria collaborative detection algorithms.

**Figure 10 sensors-23-07074-f010:**
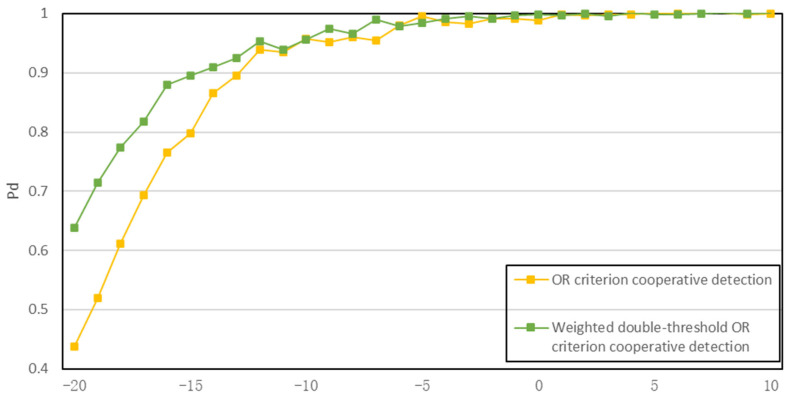
Simulation comparison diagram.

**Table 1 sensors-23-07074-t001:** Parameter table.

Parameter	Meaning
*D*	Distance between the sound source and the receiver
*X* _1_	Distance between the transmitter and the water surface
*X* _2_	Distance between the receiver and the water surface
*H*	Water depth
*T*	Transmitter
*R*	Receiver
O_1_, O_2_, O_3_	Virtual source

## Data Availability

Not applicable.

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
