# Peer review of "Spectrum-Weighted Fusion Cooperative Detection Algorithm Based on Double Thresholds for Underwater Acoustic Networks"

_sensors, 2023, doi:10.3390/s23167074_

Round 1
Reviewer 1 Report
The authors have written article on Spectrum-Weighted Fusion Cooperative Detection Algorithm Based on Double Thresholds for Underwater Acoustic Networks. The manuscript lacks novelity of research. The research article should have novel research finding. Further no proper litterature review has been done. Authors are suggested to read the previous works on Cooperative Detection Algorithm. Further the graphs are of very low quality. The quality of English writing is also very poor.
I wish you best of luck for future articles.
The quality of English writing is also very poor. It is making difficult to understand.
Reviewer 2 Report
The manuscript is written in a careless and sloppy manner. Several problems are identified and should be addressed prior to any further consideration for publication.
Citation Issues: There are inconsistencies and ambiguities in the citation styles. For instance, in line 55 "fading channels 5, low sample size6, and noise uncertainty7. Fadel F." and line 64, line 66.
Quality of Figures: The quality of the figures, specifically figures 1 and 6, is inadequate and they need to be improved for better clarity and understanding.
Unexplained Acronyms and Terms: In line 152, the term "TL22" is used without any prior explanation. It is critical to explain all terms and acronyms when they are first introduced.
Sentence Structure: The manuscript contains sentences that are poorly structured or confusing, such as in lines 157, 160, 161, 165, 167, and 169.
For example, the sentence in line 167 "Figure 1 shows the planar surface of the speed of sound, with the depth of the ocean changes, the speed of sound in the propagation process, by different degrees of influence" is hard to interpret. It appears to lack coherence and the intended meaning is unclear.
Similarly, the sentence in line 169 "Using the sound channel axis as the depth where the transmitting end is located, the sound line trajectory diagram is made as shown in Figure 2" is rather convoluted and its meaning is hard to decipher.
Issues with Figures: There are strange symbols following each parameter in figures 3, 4, and 5. Additionally, the arrows in figure 3 are not accurately placed.
Reference to Figure: In line 206, the text states “The basic detection model is shown below” - the reference should be to the specific figure number.
Equations: Equations 8 and 9 represent the received signal inconsistently as Y(t) and x(t) respectively. The representation of variables and equations, up to equation 13, should be unified and commented on for clarity.
In line 240, there is a mismatch with "pm" mentioned in the text and equation 16.
In its current form, this manuscript is not fit for publication. The issues highlighted above, and similar ones, need to be thoroughly corrected and explained.
na
Round 2
Reviewer 1 Report
The authors have worked on the previous comments. However still there are some modifications required.
1. The introduction still does not have key papers. and Literature review is missing. I would suggest to add more key papers and following paper.
Symmetric connectivity of underwater acoustic sensor networks based on multi-modal directional transducer
G Qiao, Q Liu, S Liu, B Muhammad, M Wen Sensors 21 (19), 6548 Multi‐user underwater acoustic communication using binary phase‐coded hyperbolic frequency‐modulated signals HH Zuberi, S Liu, MZ Sohail, C Pan IET Communications 16 (12), 1415-1427 Further kindly elaborate the findings of figure 8-10 as it is the key research of this research. Also the quality of figure 8-10 is very low redraw it.
And further advise to cite more relevant papers, as literature review is not so good.
English Quality is a bit poor. I request to proof read the whole paper before final submission.
Reviewer 2 Report
Please attach an additional version of the manuscript without tracked changes. The current version is difficult to read.
Furthermore, upon initial review, it appears there are still "strange symbols following each parameter in figures 3, 4, and 5." Please address this in your revision or explain it.
-
